# Breaking the Span Assumption Yields Fast Finite-Sum Minimization[*]

Robert Hannah[1][†], Yanli Liu[1][‡], Daniel O'Connor[2][§], and Wotao Yin[1][¶]

[1]*Department of Mathematics, University of California, Los Angeles*
[2]*Department of Mathematics, University of San Francisco*

## Abstract

In this paper, we show that SVRG and SARAH can be modified to be fundamentally faster than all of the other standard algorithms that minimize the sum of $n$ smooth functions, such as SAGA, SAG, SDCA, and SDCA without duality. Most finite sum algorithms follow what we call the "span assumption": Their updates are in the span of a sequence of component gradients chosen in a random IID fashion. In the big data regime, where the condition number $\kappa = \mathcal{O}(n)$, the span assumption prevents algorithms from converging to an approximate solution of accuracy $\epsilon$ in less than $n \ln(1/\epsilon)$ iterations. SVRG and SARAH do not follow the span assumption since they are updated with a hybrid of full-gradient and component-gradient information. We show that because of this, they can be up to $\Omega(1 + (\ln(n/\kappa))_+)$ times faster. In particular, to obtain an accuracy $\epsilon = 1/n^\alpha$ for $\kappa = n^\beta$ and $\alpha, \beta \in (0, 1)$, modified SVRG requires $\mathcal{O}(n)$ iterations, whereas algorithms that follow the span assumption require $\mathcal{O}(n \ln(n))$ iterations. Moreover, we present lower bound results that show this speedup is optimal, and provide analysis to help explain why this speedup exists. With the understanding that the span assumption is a point of weakness of finite sum algorithms, future work may purposefully exploit this to yield faster algorithms in the big data regime.

## 1 Introduction

Finite sum minimization is an important class of optimization problem that appears in many applications in machine learning and other areas. We consider the problem of finding an approximation $\hat{x}$ to the minimizer $x^*$ of functions $F : \mathbb{R}^d \to \mathbb{R}$ of the form:

$$F(x) = f(x) + \psi(x) \triangleq \frac{1}{n} \sum_{i=1}^{n} f_i(x) + \psi(x). \tag{1.1}$$

We assume each function $f_i$ is smooth[6], and possibly nonconvex; $\psi$ is proper, closed, and convex; and the sum $F$ is strongly convex and smooth. It has become well-known that under a variety of assumptions, functions of this form can be minimized much faster with variance

---

[*]This work was supported in part by grants: AFOSR MURI FA9550-18-1-0502, NSF DMS-1720237, and ONR N000141712162.

[†]Corresponding author: RobertHannah89@gmail.com

[‡]yanli@math.ucla.edu

[§]daniel.v.oconnor@gmail.com

[¶]WotaoYin@math.ucla.edu

[6]A function $f$ is $L$-smooth if it has an $L$-Lipschitz gradient $\nabla f$

reduction (VR) algorithms that specifically exploit the finite-sum structure. When each $f_i$ is $\mu$-strongly convex and $L$-smooth, and $\psi = 0$, SAGA [1], SAG [2], Finito/Miso [3], [4], SVRG [5], SARAH [6], SDCA [7], and SDCA without duality [8] can find a vector $\hat{x}$ with expected suboptimality $\mathbb{E}(f(\hat{x}) - f(x^*)) = \mathcal{O}(\epsilon)$ with only $\mathcal{O}((n + L/\mu)\ln(1/\epsilon))$ calculations of component gradients $\nabla f_i(x)$. This can be up to $n$ times faster than (full) gradient descent, which takes $\mathcal{O}(nL/\mu\ln(1/\epsilon))$ gradients. These algorithms exhibit sublinear convergence for non-strongly convex problems[7]. Various results also exist for nonzero convex $\psi$.

Accelerated VR algorithms have also been proposed. Katyusha [9] is a primal-only Nesterov-accelerated VR algorithm that uses only component gradients. It is based on SVRG and has complexity $\mathcal{O}((n + \sqrt{n\kappa})\ln(1/\epsilon)))$ for condition number $\kappa$ which is defined as $L/\mu$. In [10], the author devises an accelerated SAGA algorithm that attains the same complexity using component proximal steps. In [11], the author devises an accelerated primal-dual VR algorithm. There also exist "catalyst" [12] accelerated methods [13], [14]. However, catalyst methods appear to have a logarithmic complexity penalty over Nesterov-accelerated methods, a defect that researchers have been able to correct.

In [15], authors show that a class of algorithms that includes SAGA, SAG, Finito (with replacement), Miso, SDCA without duality, etc. have complexity $K(\epsilon)$ lower bounded by $\Omega((n + \sqrt{n\kappa})\ln(1/\epsilon))$ for problem dimension $d \geq 2K(\epsilon)$. More precisely, the lower bound applies to algorithms that satisfy what we will call the **span condition**. That is

$$x^{k+1} \in x^0 + \text{span}\{\nabla f_{i_0}(x^0), \nabla f_{i_1}(x^1), \ldots, \nabla f_{i_k}(x^k)\} \tag{1.2}$$

for some fixed IID random variable $i_k$ over the indices $\{1, \ldots, n\}$. Later, [16] and [17] extend lower bound results to algorithms that do not follow the span assumption: SDCA, SVRG, SARAH, accelerated SAGA, etc.; but with a weaker lower bound of $\Omega(n + \sqrt{n\kappa}\ln(1/\epsilon))$. The difference in these two expressions was thought to be a proof artifact that would later be resolved.

However we show a surprising result in Section 2, that SVRG, and SARAH can be fundamentally faster than methods that satisfy the span assumption, with the full gradient steps playing a critical role in their speedup. More precisely, for $\kappa = \mathcal{O}(n)$, SVRG and SARAH can be modified to reach an accuracy of $\epsilon$ in $\mathcal{O}((\frac{n}{1+(\ln(n/\kappa))_+})\ln(1/\epsilon))$ gradient calculations[8], instead of the $\Theta(n\ln(1/\epsilon))$ iterations required for algorithms that follow the span condition.

We also improve the lower bound of [17] to $\Omega(n + (\frac{n}{1+(\ln(n/\kappa))_+} + \sqrt{n\kappa})\ln(1/\epsilon))$ in Section 2.1. That is, we show that the complexity $K(\epsilon)$ of a very general class of algorithm that includes all of the above algorithms satisfies the lower bound:

$$K(\epsilon) = \begin{cases} \Omega(n + \sqrt{n\kappa}\ln(1/\epsilon)), & \text{for } n = \mathcal{O}(\kappa), \\ \Omega(n + \frac{n}{1+(\ln(n/\kappa))_+}\ln(1/\epsilon)), & \text{for } \kappa = \mathcal{O}(n). \end{cases} \tag{1.3}$$

Hence when $\kappa = \mathcal{O}(n)$ our modified SVRG has optimal complexity, and when $n = \mathcal{O}(\kappa)$, Katyusha is optimal.

SDCA doesn't quite follow the span assumption. Also the dimension $n$ of the dual space on which the algorithm runs is inherently small in comparison to $k$, the number of iterations. We complete the picture using different arguments, by showing that its complexity is lower bounded by $\Omega(n\ln(1/\epsilon))$ in Section 2.2. Hence SDCA doesn't attain this logarithmic speedup. We leave the analysis of accelerated SAGA, accelerated SDCA, and other algorithms to future work.

Our results identify a significant obstacle to high performance when $n \gg \kappa$. The speedup that SVRG and SARAH can be modified to attain in this scenario is somewhat accidental since their original purpose was to minimize memory overhead. However, with the knowledge that this assumption is a point of weakness for VR algorithms, future work may more purposefully exploit this to yield better speedups than SVRG and SARAH can currently attain. Though the complexity of SVRG and SARAH can be made optimal to within a

constant factor, this factor is somewhat large, and could potentially be reduced substantially. Though it is unclear how much of a speedup is possible.

Having $n \gg \kappa$, which has been referred to as the "big data condition", is rather common, especially in regularized empirical risk minimization (ERM). For instance [2] remarks that $\kappa = \sqrt{n}$ is a nearly optimal choice for regularization for empirical risk minimization in the low dimensional setting. In the high-dimensional setting, the authors of [2] claim there is no analysis that they are aware *that doesn't imply* that we should set the regularization term to ensure $n = \mathcal{O}(\kappa)$. In [18], authors consider regularized ERM for minimizing a stochastic objective. They argue that the optimal choice of regularization parameter $\lambda$ corresponds to $\kappa = \sqrt{n}$. [19] considers regularized SVM with $\kappa = n^{\beta}$ for $\beta < 1$.

Hence our results have wide application. In the settings described above, we have the following corollary that implies a complexity improvement from $\mathcal{O}(n \ln(n))$ to $\mathcal{O}(n)$. This will follow from Corollary 2 ahead.

**Corollary 1.** *To obtain accuracy $\epsilon = 1/n^{\alpha}$ for $\kappa = n^{\beta}$ and some $\alpha, \beta \in (0,1)$, modified SVRG requires $\mathcal{O}(n)$ iterations, whereas algorithms that follow the span assumption require $\mathcal{O}(n \ln(n))$ iterations [11] for sufficiently large problem dimension $d$.*

For large-scale problems, this $\ln(n)$ factor can be rather large: For instance in the KDD Cup 2012 dataset ($n = 149,639,105$ and $\ln(n) \approx 18$), Criteo's Terabyte Click Logs ($n = 4,195,197,692$ and $\ln(n) \approx 22$), etc. Non-public internal company datasets can be far larger, with $n$ potentially larger than $10^{15}$.

We also analyze Prox-SVRG in the case where $f_i$ are smooth and potentially nonconvex, but the sum $F$ is strongly convex. We build on the work of [20], which proves state-of-the-art complexity bounds for this setting, and show that we can attain a similar logarithmic speedup without modification. Lower bounds for this context are lacking, so it is unclear if this result can be further improved.

## 2   Optimal Convex SVRG

In this section, we show that the Prox-SVRG algorithm proposed in [21] for problem (1.1) can be sped up by a factor of $\Omega(1 + (\ln(n/\kappa))_+)$ when $\kappa = \mathcal{O}(n)$. A similar speedup is clearly possible for vanilla SVRG and SARAH, which have similar rate expressions. We then refine the lower bound analysis of [17] to show that the complexity is optimal[9] when $\kappa = \mathcal{O}(n)$. Katyusha is optimal in the other scenario when $n = \mathcal{O}(\kappa)$ by [22].

**Assumption 1.**   $f_i$ is $L_i-$Lipschitz differentiable for $i = 1, 2, ..., n$. That is,

$$\|\nabla f_i(x) - \nabla f_i(y)\| \leq L_i \|x - y\| \quad \text{for all } x, y \in \mathbb{R}^d.$$

$f$ is $L-$Lipschitz differentiable. $F$ is $\mu-$strongly convex. That is,

$$F(y) \geq F(x) + \langle \tilde{\nabla} F(x), y - x \rangle + \frac{\mu}{2} \|y - x\|^2 \quad \text{for all } x, y \in \mathbb{R}^d \text{ and } \tilde{\nabla} F(x) \in \partial F(x).$$

**Assumption 2.**         $f_i$ is convex for $i = 1, 2, ..., n$; and $\psi$ is proper, closed, and convex.

**Algorithm 1** Prox-SVRG($F, x^0, \eta, m$)

---

**Input:** $F(x) = \psi(x) + \frac{1}{n}\sum_{i=1}^{n} f_i(x)$, initial vector $x^0$, step size $\eta > 0$, number of epochs $K$, probability distribution $P = \{p_1, \ldots, p_n\}$
**Output:** vector $x^K$
1:   $M^k \sim \text{Geom}(\frac{1}{m})$;
2: **for** $k \leftarrow 0, \ldots, K-1$ **do**
3:     $w_0 \leftarrow x^k$; $\mu \leftarrow \nabla f(x^k)$;
4:     **for** $t \leftarrow 0, \ldots, M^k$ **do**
5:       pick $i_t \in \{1, 2, \ldots, n\} \sim P$ randomly;
6:       $\tilde{\nabla}_t = \mu + \left(\nabla f_{i_t}(w_t) - \nabla f_{i_t}(w_0)\right)/(np_{i_t})$;
7:       $w_{t+1} = \arg\min_{y \in \mathbb{R}^d}\{\psi(y) + \frac{1}{2\eta}\|y - w_t\|^2 + \langle\tilde{\nabla}_t, y\rangle\}$;
8:     **end for**
9:     $x^{k+1} \leftarrow w_{M+1}$;
10: **end for**

---

We make Assumption 1 throughout the paper, and Assumption 2 in this section. Recall the Prox-SVRG algorithm of [21], which we reproduce in Algorithm 1. The algorithm is organized into a series of $K$ **epochs** of size $M^k$, where $M^k$ is a geometric random variable with success probability $1/m$. Hence epochs have an expected length of $m$. At the start of each epoch, a snapshot $\mu = \nabla f(x^k)$ of the gradient is taken. Then for $M^k$ steps, a random component gradient $\nabla_{i_t} f(w_t)$ is calculated, for an IID random variable $i_t$ with fixed distribution $P$ given by $\mathbb{P}[i_t = i] = p_i$. This component gradient is used to calculate an unbiased estimate $\tilde{\nabla}_t$ of the true gradient $\nabla f(w_t)$. Each time, this estimate is then used to perform a proximal-gradient-like step with step size $\eta$. At the end of these $M^k$ steps, a new epoch of size $M^{k+1}$ is started, and the process continues.

We first recall a modified Theorem 1 from [21]. The difference is that in [21], the authors used an epoch length of $m$, whereas we use a random epoch length $M^k$ with expectation $\mathbb{E}M^k = m$. The proof and theorem statement only require only trivial modifications to account for this. This modification is only to unify the different version of SVRG in [21] and [20], and makes no difference to the result.

It becomes useful to define the **effective Lipschitz constant** $L_Q = \max_i L_i/(p_i n)$, and the **effective condition number** $\kappa_Q = L_Q/\mu$ for this algorithm. These reduce to the standard Lipschitz constant $L$, and the standard condition number $\kappa$ in the standard uniform scenario where $L_i = L, \forall i$, and $P$ is uniform.

**Theorem 1. Complexity of Prox-SVRG.** Let Assumptions 1 and 2 hold. Then Prox-SVRG defined in Algorithm 1 satisfies

$$\mathbb{E}[F(x^k) - F(x^*)] \leq \rho^k[F(x^0) - F(x^*)] \quad (2.1) \qquad \text{for } \rho = \frac{1 + \mu\eta(1 + 4mL_Q\eta)}{\mu\eta m(1 - 4L_Q\eta)}. \quad (2.2)$$

In previous work, the optimal parameters were not really explored in much detail. In the original paper [5], the author suggest $\eta = 0.1/L$, which results in linear convergence rate $1/4 \leq \rho \leq 1/2$ for $m \geq 50\kappa$. In [21], authors also suggest $\eta = 0.1/L$ for $m = 100\kappa$, which yields $\rho \approx 5/6$. However, they observe that $\eta = 0.01/L$ works nearly as well. In [6], authors obtain a similar rate expression for SARAH and suggest $\eta = 0.5/L$ and $m = 4.5\kappa$ which yields $\rho \approx 7/9$. In the following corollary, we propose a choice of $\eta$ and $m$ that leads to an optimal complexity to within a constant factor for $\kappa = \mathcal{O}(n)$. This result helps explain why the optimal step size observed in prior work appears to be much smaller than the "standard" gradient descent step of $1/L$.

**Corollary 2.** *Let the conditions of Theorem 1 hold, and let $m = n + 121\kappa_Q$, and $\eta = \kappa_Q^{\frac{1}{2}} m^{-\frac{1}{2}}/(2L_Q)$. The Prox-SVRG in Algorithm 1 has convergence rate $\rho \leq \sqrt{\frac{100}{121 + (n/\kappa_Q)}}$, and hence it needs:*

$$K(\epsilon) = \mathcal{O}\left(\left(\frac{n}{1 + (\ln(\frac{n}{\kappa_Q}))_+} + \kappa_Q\right)\ln\frac{1}{\epsilon} + n + \kappa_Q\right) \tag{2.3}$$

*iterations in expectation to obtain a point $x^{K(\epsilon)}$ such that $\mathbb{E}\big[f\big(x^{K(\epsilon)}\big) - f(x^*)\big] < \epsilon$.*

This result is proven in Appendix A. The $n + \kappa_Q$ term is needed because we assume that at least one epoch is completed. For $n = \mathcal{O}(\kappa_Q)$, we have a similar convergence rate ($\rho \approx \frac{10}{11}$) and complexity to algorithms that follow the span assumption. For $n \gg \kappa_Q$, we have a convergence rate $\rho \sim \sqrt{\kappa_Q/n} \to 0$, and complexity $\mathcal{O}\Big(\frac{n}{1+(\ln(n/\kappa))}\ln(1/\epsilon)\Big)$, which can can be much better than $n\ln(1/\epsilon)$. See also Corollary 1. The corresponding result and proof for SARAH is nearly identical, and we do not include this.

In order to obtain this speedup, some estimate of the condition number must be known. However this is often not a problem. In the case of a regularization term for empirical risk minimization, the strong convexity modulus is hand-picked based on the number of examples $n$. In other cases, we can simply tune an estimate of the parameter $\kappa$ with the assurance that this can yield a logarithmic speedup.

**Remark 1.** In Theorem 1 and Corollary 2, the optimal choice of the probability distribution $P = \{p_1, p_2, ..., p_n\}$ on $\{1, 2, ..., n\}$ is $p_i = \frac{L_i}{\sum_{i=1}^{n} L_j}$ for $i = 1, 2, ..., n$, and $L_Q = \frac{\sum_{i=1}^{n} L_i}{n}$.

## 2.1 Optimality

The major difference between SAGA, SAG, Miso/Finito, and SDCA without duality, and SVRG and SARAH, is that the former satisfy what we call the **span condition** (1.2). SVRG, and SARAH, do not, since they also involve full-gradient steps. We refer to SVRG, and SARAH as **hybrid methods**, since they use full-gradient and partial gradient information to calculate their iterations. We assume for simplicity that $L_i = L$, for all $i$, and that $\psi = 0$. We now present a rewording of Corollary 3 from [11].

**Corollary 3.** *For every $\epsilon$ and randomized algorithm on (1.1) that follows the span assumption, there are a dimension $d$, and $L$-smooth, $\mu$-strongly convex functions $f_i$ on $\mathbb{R}^d$ such that the algorithm takes at least $\Omega((n + \sqrt{\kappa n})\ln(1/\epsilon))$ steps to reach sub-optimality $\mathbb{E}f\big(x^k\big) - f(x^*) < \epsilon$.*

The above algorithms that satisfy the span condition all have known upper complexity bounds of $\mathcal{O}((n + \kappa)\ln(1/\epsilon))$, and hence for $\kappa = \mathcal{O}(n)$ we have a sharp convergence rate.

However, it turns out that the span assumption is an obstacle to faster convergence when $n \gg \kappa$ (at least for sufficiently high dimension). In the following theorem, we improve[10] the analysis of [17], to show that the complexity of SVRG obtained in Corollary 2 is optimal to within a constant factor without fundamentally different assumptions on the class of algorithms that are allowed. Clearly this also applies to SARAH. The theorem is actually far more general, and applies to a general class of algorithms called $p-CLI$ *oblivious* algorithms introduced in [17]. This class contains all VR algorithms mentioned in this paper. In Appendix B, we recall the definition of $p-$CLI oblivious algorithms, as well as providing the proof of a more general version of Theorem 2.

**Theorem 2. Lower complexity bound of Prox-SVRG and SARAH.** For all $\mu, L$, there exist $L$-smooth, and $\mu$-strongly convex functions $f_i$ such that at least[11]

$$K(\epsilon) = \tilde{\Omega}\bigg(\bigg(\frac{n}{1 + (\ln(\frac{n}{\kappa}))_+} + \sqrt{n\kappa}\bigg)\ln\frac{1}{\epsilon} + n\bigg) \tag{2.4}$$

iterations are needed for SVRG or SARAH to obtain expected suboptimality $\mathbb{E}[f(K(\epsilon)) - f(x^*)] < \epsilon$.

## 2.2 SDCA

To complete the picture, in the following proposition, which we prove in Appendix C, we show that SDCA has a complexity lower bound of $\Omega(n\ln(1/\epsilon))$. Hence it attains no logarithmic

speedup. SDCA aims to solve the following problem:

$$\min_{x \in \mathbb{R}^d} F(x) = \frac{1}{n} \sum_{i=1}^{n} f_i(x) = \frac{1}{n} \sum_{i=1}^{n} \left( \phi_i(x^T y_i) + \frac{\lambda}{2} \|x\|^2 \right),$$

where each $y_i \in \mathbb{R}^d$, $\phi_i : \mathbb{R} \to \mathbb{R}$ is convex and smooth. It does so with coordinate minimization steps on the corresponding dual problem:

$$\min_{\alpha \in \mathbb{R}^n} D(\alpha) := \frac{1}{n} \sum_{i=1}^{n} \phi_i^*(-\alpha_i) + \frac{\lambda}{2} \left\| \frac{1}{\lambda n} \sum_{i=1}^{n} \alpha_i y_i \right\|^2,$$

Here $\phi_i^*(u) := \max_z \left( zu - \phi_i(z) \right)$ is the convex conjugate of $\phi_i$. Let $i_k$ be an IID sequence of uniform random variables on $\{1, ..., n\}$. SDCA updates a dual point $\alpha^k$, while maintaining a corresponding primal vector $x^k$. SDCA can be written as:

$$\alpha_i^{k+1} = \begin{cases} \alpha_i^k, & \text{if } i \neq i_k, \\ \arg\min_z D(\alpha_1^k, ..., \alpha_{i-1}^k, z, \alpha_{i+1}^k, ..., \alpha_n^k), & \text{if } i = i_k, \end{cases} \tag{2.5}$$

$$x^{k+1} = \frac{1}{n\lambda} \sum_{i=1}^{n} \alpha_i^{k+1} y_i, \tag{2.6}$$

Since SDCA doesn't follow the span assumption, and the number of iterations $k$ is much greater than the dual problem dimension $n$, different arguments to the ones used in [11] must be used. Motivated by the analysis in [23], which only proves a lower bound for dual suboptimality, we have the following lower complexity bound, which matches the upper complexity bound given in [7] for $\kappa = \mathcal{O}(n)$.

**Proposition 1. Lower complexity bound of SDCA.** *For all $\mu, L, n > 2$, there exist $n$ functions $f_i$ that are $L-$smooth, and $\mu-$strongly convex such that*

$$K(\epsilon) = \Omega\left(n \ln \frac{1}{\epsilon}\right) \tag{2.7}$$

*iterations are needed for SDCA to obtain expected suboptimality $\mathbb{E}[F(x^{K(\epsilon)}) - F(x^*)] \leq \epsilon$.*

## 3  Why are hybrid methods faster?

In this section, we explain why SVRG and SARAH, which are a hybrid between full-gradient and VR methods, are fundamentally faster than other VR algorithms. We consider the performance of these algorithms on a variation of the adversarial function example from [11], [24]. The key insight is that the span condition makes this adversarial example hard to minimize, but that the full gradient steps of SVRG and SARAH make it easy when $n \gg \kappa$.

We conduct the analysis in $\ell^2$, for simplicity[12], since the argument readily applies to $\mathbb{R}^d$. Consider the function introduced in [24] that we introduce for the case $n = 1$:

$$\phi(x) = \frac{L - \sigma}{4} \left( \frac{1}{2} \langle x, Ax \rangle - \langle e_1, x \rangle \right), \text{ for } A = \begin{pmatrix} 2 & -1 & & & \\ -1 & 2 & -1 & & \\ & -1 & 2 & \ddots & \\ & & \ddots & \ddots & \end{pmatrix}$$

The function $\phi(x) + \frac{1}{2}\sigma\|x\|^2$ is $L$-smooth and $\sigma$-strongly convex. Its minimizer $x^*$ is given by $\left(q_1, q_1^2, q_1^3, \ldots\right)$ for $q_1 = \left(\kappa^{1/2} - 1\right)/\left(\kappa^{1/2} + 1\right)$. We assume that $x^0 = 0$ with no loss in generality. Let $N(x)$ be position of the last nonzero in the vector. E.g. $N(0, 2, 3, 0, 4, 0, 0, 0, \ldots) = 5$. $N(x)$ is a control on how close $x$ can be to the solution. If $N(x) = N$, then clearly:

$$\|x - x^*\|^2 \geq \min_{y \text{ s.t. } N(y) = N} \|y - x^*\|^2 = \left\| \left(0, \ldots, 0, q_1^{N+1}, q_1^{N+2}, \ldots\right) \right\|^2 = q_1^{2N+2}/\left(1 - q_1^2\right)$$

Because of the tridiagonal pattern of nonzeros in the Hessian $\nabla_x^2\left(\phi(x) + \frac{1}{2}\sigma\|x\|^2\right)(y) = \frac{L-\sigma}{4}A + \sigma I$, the last nonzero $N(x^k)$ of $x^k$ can only increase by 1 per iteration *by any algorithm that satisfies that span condition* (e.g. gradient descent, accelerated gradient descent, etc.). Hence since we have $N(x^0) = 0$, we have $\|x^k - x^*\|^2 / \|x^0 - x^*\|^2 \geq q_1^{2k}$.

For the case $n > 1$, let the solution vector $x = (x_1, \ldots, x_n)$ be split into $n$ coordinate blocks, and hence define:

$$f(x) = \sum_{i=1}^n \left(\phi(x_i) + \frac{1}{2}\sigma\|x\|^2\right) \tag{3.1}$$

$$= \sum_{i=1}^n \left(\frac{L-\sigma}{4}\left(\frac{1}{2}\langle x_i, Ax_i\rangle - \langle e_1, x_i\rangle\right) + \frac{1}{2}(\sigma n)\|x_i\|^2\right)$$

$$= \sum_{i=1}^n \left(\frac{(L-\sigma+\sigma n) - \sigma n}{4}\left(\frac{1}{2}\langle x_i, Ax_i\rangle - \langle e_1, x_i\rangle\right) + \frac{1}{2}(\sigma n)\|x_i\|^2\right). \tag{3.2}$$

$f$ is clearly the sum of $n$ convex $L$-smooth functions $\phi(x_i) + \frac{1}{2}\sigma\|x\|^2$, that are $\sigma$-strongly convex. (3.2) shows it is $\sigma n$-strongly convex and $L - \sigma + \sigma n$-smooth with respect to coordinate $x_i$. Hence the minimizer is given by $x_i = \left(q_n, q_n^2, q_n^3, \ldots\right)$ for $q_n = \left(\left(\frac{\kappa-1}{n}+1\right)^{1/2} - 1\right)/\left(\left(\frac{\kappa-1}{n}+1\right)^{1/2} + 1\right)$ for all $i$. Similar to before, $(N(x_1), \ldots, N(x_n))$ controls how close $x$ can be to $x^*$:

$$\frac{\|x - x^*\|^2}{\|x^*\|^2} = \frac{\sum_{i=1}^n \|x_i - (q_n, q_n^2, \ldots)\|^2}{nq_n^2/(1 - q_n^2)} \geq \sum_{i=1}^n q_n^{2N(x_i)}/n$$

Let $I_{K,i}$ be the number of times that $i_k = i$ for $k = 0, 1, \ldots, K - 1$. For algorithms that satisfy the span assumption, we have $N(x_i^k) \leq I_{k,i}$. If we assume that $i_k$ is uniform, then $I_{K,i}$ is a binomial random variable of probability $1/n$ and size $k$. Hence:

$$\mathbb{E}\|x^k - x^*\|^2/\|x^0 - x^*\|^2 \geq \mathbb{E}\sum_{i=1}^n q_n^{2N(x_i^k)}/n \geq \mathbb{E}\sum_{i=1}^n q_n^{2I_{k,i}}/n$$

$$= \mathbb{E}q_n^{2I_{k,i}} = \left(1 - n^{-1}(1 - q_n^2)\right)^k \tag{3.3}$$

$$\geq \left(1 - 4n^{-1}/\left(\left(\frac{\kappa-1}{n}+1\right)^{1/2} + 1\right)\right)^k$$

$$\geq \left(1 - 2n^{-1}\right)^k$$

for $n \geq \kappa$. the second equality in (3.3) follows from the fact that $I_{i,k}$ is a binomial random variable. Hence after 1 epoch, $\mathbb{E}\|x^k - x^*\|^2$ decreases by a factor of at most $\approx e^2$, whereas for SVRG it decreases by at least a factor of $\sim (n/\kappa)^{1/2}$, which is $\gg e^2$ for $n \gg \kappa$. To help understand why, consider trying the above analysis on SVRG for 1 epoch of size $n$. Because of the full-gradient step, we actually have $N(w_i^n) \leq 1 + I_{n,i}$, and hence:

$$\mathbb{E}\|w^n - x^*\|^2/\|x^0 - x^*\|^2 \geq \mathbb{E}\sum_{i=1}^n q_n^{2(I_{n,1}+1)} \geq q_n^2\left(1 - 2n^{-1}\right)^n \approx \left(\frac{1}{4}\frac{\kappa-1}{n}\right)^2 e^{-2}$$

Hence attempting the above results in a much weaker lower bound.

What it comes down to is that when $n \gg \kappa$, we have $\mathbb{E}\sum_{i=1}^n q_n^{2I_{i,k}}/n \gg \mathbb{E}\sum_{i=1}^n q_n^{2\mathbb{E}I_{i,k}}/n$. The interpretation is that for this objective, the progress towards a solution is limited by the component function $f_i$ that is minimized the least. The full gradient step ensures that at least some progress is made toward minimizing every $f_i$. For algorithms that follow the span assumption, there will invariably be many indices $i$ for which no gradient $\nabla f_i$ is calculated, and hence $x_i^k$ can make no progress towards the minimum. This may be related to the observation that sampling without replacement can often speed up randomized

algorithms. However, on the other hand, it is well known that full gradient methods fail to achieve a good convergence rate for other objectives with the same parameters $\mu, L, n$ (e.g. $f(x) = \frac{1}{n}\sum_{i=1}^{n}\phi(x) + \frac{1}{2}\mu\|x\|^2$). Hence we conclude that it is because SVRG is a hybrid method, that combines both full-gradient and VR elements, that it is able to outperform both VR and full-gradient algorithms.

## 4  Prox-SVRG for strongly convex sum of smooth nonconvex function

In this section we show that this logarithmic speedup is still possible if we relax Assumption 2: that each $f_i$ is convex. By Assumption 1, the functions $f_i$ are still smooth, though possibly nonconvex. The sum $F$ though is strongly convex, and smooth. This is based on the analysis of Prox-SVRG found in [20]. The proof of Theorem 3 can be found in Appendix D.

**Theorem 3.**  Under Assumption 1, let $x^* = \arg\min_x F(x)$, $\overline{L} = (\sum_{i=1}^{n}\frac{L_i^2}{n^2 p_i})^{\frac{1}{2}}$, $\kappa = \frac{L}{\mu}$, and $\eta = \frac{1}{2}\min\{\frac{1}{L}, (\frac{1}{\overline{L}^2 m})^{\frac{1}{2}}\}$. Then the Prox-SVRG in Algorithm 1 satisfies

$$\mathbb{E}[F(x^k) - F(x^*)] \le \mathcal{O}(\rho^k)[F(x^0) - F(x^*)], \quad (4.1) \qquad \text{for } \rho = \frac{1}{1 + \frac{1}{2}m\eta\mu}. \quad (4.2)$$

Hence for $m = \min\{n, 2\}$, in order to obtain an $\epsilon$-optimal solution in terms of function value, the SVRG in Algorithm 1 needs at most

$$K = \mathcal{O}\Big(\big(\frac{n}{\ln\left(1 + \frac{n}{4\kappa}\right)} + \frac{n}{\ln(1 + (\frac{n\mu^2}{4\overline{L}^2})^{1/2})} + \kappa + \sqrt{n}\frac{\overline{L}}{\mu}\big)\ln\frac{1}{\epsilon}\Big) + 2n \quad (4.3)$$

gradient evaluations in expectation.

The complexity of nonconvex SVRG using the original analysis of [9] would have been

$$K = \mathcal{O}\big((n + \kappa + \sqrt{n}\frac{\overline{L}}{\mu})\ln\frac{1}{\epsilon}\big) \quad (4.4)$$

Hence we have obtained a similar logarithmic speedup as we obtained in Corollary 2. There are no known nontrivial lower bounds in this regime, and so it is not immediately clear whether our complexity is optimal.

**Remark 2.**  In Theorem 3, the optimal choice of the probability distribution $P = \{p_1, p_2, ..., p_n\}$ on $\{1, 2, ..., n\}$ is $p_i = \frac{L_i^2}{\sum_{i=1}^{n}L_j^2}$ for $i = 1, 2, ..., n$, and $\overline{L} = (\frac{\sum_{i=1}^{n}L_i^2}{n})^{\frac{1}{2}}$.

## 5  Experiments

In this section we compare the performance of SVRG, and SARAH to SAGA to verify our conclusions. We solve the regularized least squares problem

$$\text{minimize} \quad \frac{1}{2n}\|Ax - b\|_2^2 + \frac{\lambda}{2}\|x\|_2^2. \quad (5.1)$$

The matrix $A$ and vector $b$ are generated randomly with entries uniformly distributed between 0 and 1. In this experiment, $A$ has $n = 16000$ rows and 20 columns. The parameter $\lambda$ is chosen to control the condition number $\kappa = L/\mu$ of the problem. Figure 5.1 compares SAGA, SVRG, and SARAH for three instances of problem (5.1) with conditions numbers $\kappa = 5, \kappa = 10$, and $\kappa = 20$. In order to provide a fair comparison, step sizes were tuned individually for each algorithm and each problem instance.

There does appear to be a small but noticeable effect when $n/\kappa$ is large. In all our experiments, SAGA appears to converge very quickly initially, but slows significantly after a few iterations. To compensate for this, we compare the convergence speed of the three algorithms *after the first few iterations* in terms of decibels per epoch[13]. For $\kappa = 5$, we obtain convergence speeds

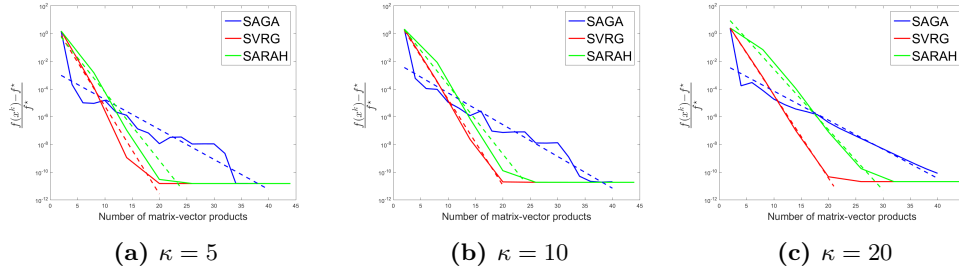

**(a)** $\kappa = 5$  **(b)** $\kappa = 10$  **(c)** $\kappa = 20$

**Figure 5.1:** Comparison of SAGA, SVRG, and SARAH for various values of the condition number $\kappa$.

of 2.1, 4.9, 6.3 decibels/epoch for SAGA, SARAH, and SVRG respectively. For $\kappa = 10$ these values are 2.3, 5.0, and 6.2 respectively; and for $\kappa = 20$ these values are 2.1, 4.3, and 6.0. So SVRG converges above $3\times$ faster than SAGA in the long term, even though SVRG iterations are twice as expensive as SAGA's.

It is not yet clear whether this effect will have practical impact. However we see a few obvious future directions. Firstly, SVRG and SARAH were never intentionally designed to exploit this logarithmic speedup. It's possible that designing an algorithm with this in mind will yield greater speedup. Secondly, it should be investigating whether the initial speed burst of SAGA can be incorporated into an SVRG-like algorithm. This will make it more competitive. Thirdly, SVRG and SARAH have iterations that are twice as expensive as SAGA's because of the full gradient steps. It should be investigated whether there is a way of retaining this logarithmic speedup while reducing the iteration cost. Perhaps large batch gradients instead of full gradient will be sufficient to yield this speedup, and avoid the high cost of full gradient steps.

## Footnotes

[7]SDCA must be modified however with a dummy regularizer.

[8]We define $(a)_+$ as $\max\{a, 0\}$ for $a \in \mathbb{R}$.

[9]I.e. the complexity cannot be improved among a very broad class of finite-sum algorithms.

[10]Specifically, we improve the analysis of Theorem 2 from this paper.

[11]We absorb some smaller low-accuracy terms (high $\epsilon$) as is common practice. Exact lower bound expressions appear in the proof.

[12]This is the Hilbert space of sequences $(x_i)_{i=1}^{\infty}$ with $\sum_{i=1}^{\infty} x_i^2 < \infty$

[13]Decibels are a logarithmic scale. 10 decibels corresponds to a 10-fold increase, 100 decibels corresponds to a 100-fold increase. This is the natural way to compare speeds for linearly converging error.

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
