[Supplementary Material]

# A  Upper Complexity Bound for Convex SVRG

*Proof of Theorem 1 and Corollary 2.* (2.1) and (2.2) follows directly from the analysis of [21, Thm 3.1] with slight modification.

For the linear rate $\rho$ in (2.2), we have

$$
\begin{aligned}
\rho &\overset{(a)}{\leq} 2\left(\frac{1}{\mu\eta m} + 4L_Q\eta + \frac{1}{m}\right) \\
&\overset{(b)}{=} 2\left(\frac{1}{\mu\eta m} + 2\kappa_Q^{\frac{1}{2}}m^{-\frac{1}{2}}\right) + \frac{2}{m} \\
&\overset{(c)}{=} 2\left(\frac{1}{\mu m}2L_Q\kappa_Q^{-\frac{1}{2}}m^{\frac{1}{2}} + 2\kappa_Q^{\frac{1}{2}}m^{-\frac{1}{2}}\right) + \frac{2}{m} \\
&= 8\kappa_Q^{\frac{1}{2}}m^{-\frac{1}{2}} + \frac{2}{m} \\
&\overset{(d)}{\leq} 8\kappa_Q^{\frac{1}{2}}m^{-\frac{1}{2}} + 2\kappa_Q^{\frac{1}{2}}m^{-\frac{1}{2}} \\
&= 10\kappa_Q^{\frac{1}{2}}m^{-\frac{1}{2}},
\end{aligned}
$$

where (a) is by $\eta = \frac{\kappa_Q^{\frac{1}{2}}m^{-\frac{1}{2}}}{2L_Q} \leq \frac{1}{22L_Q} \leq \frac{1}{8L_Q}$, (b) is by $\eta = \frac{\kappa_Q^{\frac{1}{2}}m^{-\frac{1}{2}}}{2L_Q}$, (c) is by $\frac{1}{\eta} = 2L_Q m^{\frac{1}{2}}\kappa_Q^{-\frac{1}{2}}$, and (d) follows from $\kappa_Q^{\frac{1}{2}}m^{\frac{1}{2}} \geq 1$.

Therefore, the epoch complexity (i.e. the number of epochs required to reduce the suboptimality to below $\epsilon$) is

$$
\begin{aligned}
K_0 &= \left\lceil \frac{1}{\ln(\frac{1}{10}m^{\frac{1}{2}}\kappa_Q^{-\frac{1}{2}})} \ln \frac{F(x^0) - F(x^*)}{\epsilon} \right\rceil \\
&\leq \frac{1}{\ln(\frac{1}{10}m^{\frac{1}{2}}\kappa_Q^{-\frac{1}{2}})} \ln \frac{F(x^0) - F(x^*)}{\epsilon} + 1 \\
&= \frac{2}{\ln(1.21 + \frac{1}{100}\frac{n}{\kappa_Q})} \ln \frac{F(x^0) - F(x^*)}{\epsilon} + 1 \\
&= \mathcal{O}\left(\frac{1}{\ln(1.21 + \frac{n}{100\kappa_Q})} \ln \frac{1}{\epsilon}\right) + 1
\end{aligned}
$$

where $\lceil \cdot \rceil$ is the ceiling function, and the second equality is due to $m = n + 121\kappa_Q$.

Hence, the gradient complexity is

$$
\begin{aligned}
K &= (n + m)K_0 \\
&\leq \mathcal{O}\left(\frac{n + \kappa_Q}{\ln(1.21 + \frac{n}{100\kappa_Q})} \ln \frac{1}{\epsilon}\right) + n + 121\kappa_Q,
\end{aligned}
$$

which is equivalent to (2.3). □

# B  Lower Complexity Bound for Convex SVRG

**Definition 2.** [17, Def. 2] An optimization algorithm is called a Canonical Linear Iterative (CLI) optimization algorithm, if given a function $F$ and initialization points $\{w_i^0\}_{i \in J}$, where $J$ is some index set, it operates by iteratively generating points such that for any $i \in J$,

$$
w_i^{k+1} = \sum_{j \in J} O_F(w_j^k; \theta_{ij}^k), \quad k = 0, 1, \ldots
$$

holds, where $\theta_{ij}^k$ are parameters chosen, stochastically or deterministically, by the algorithm, possibly depending on the side-information. $O_F$ is an oracle parameterized by $\theta_{ij}^k$. If the

parameters do not depend on previously acquired oracle answers, we say that the given algorithm is oblivious. Lastly, algorithms with $|J| \leq p$, for some $p \in \mathbb{N}$, are denoted by p-CLI.

In [17], two types of oblivious oracles are considered. The generalized first order oracle for $F(x) = \frac{1}{n}\sum_{i=1}^{n} f_i(x)$

$$O(w; A, B, C, j) = A\nabla f_j(w) + Bw + C, \quad A, B \in \mathbb{R}^{d \times d}, C \in \mathbb{R}^d, j \in [n].$$

The steepest coordinate descent oracle for $F(x) = \frac{1}{n}\sum_{i=1}^{n} f_i(x)$ is given by

$$O(w; i, j) = w + t^* e_i, \quad t^* \in \arg\min_{t \in \mathbb{R}} f_j(w_1, ..., w_{i-1}, w + t, w_{i+1}, ..., w_d), j \in [n],$$

where $e_i$ is the $i$th unit vector. SDCA, SAG, SAGA, SVRG, SARAH, etc. without proximal terms are all $p-$CLI oblivious algorithms.

We now state the full version of Theorem 2.

**Theorem 4. Lower complexity bound oblivious p-CLI algorithms.** For any oblivious p-CLI algorithm $A$, for all $\mu, L, k$, there exist $L$-smooth, and $\mu$-strongly convex functions $f_i$ such that at least[14]:

$$K(\epsilon) = \tilde{\Omega}\left(\left(\frac{n}{1 + (\ln(\frac{n}{\kappa}))_+} + \sqrt{n\kappa}\right)\ln\frac{1}{\epsilon} + n\right) \tag{B.1}$$

iterations are needed for $A$ to obtain expected suboptimality $\mathbb{E}[f(K(\epsilon)) - f(X^*)] < \epsilon$.

*Proof of Theorem 4.* In this proof, we use lower bound given in [17, Thm 2], and refine its proof for the case $n \geq \frac{1}{3}\kappa$.

[17, Thm 2] gives the following lower bound,

$$K(\epsilon) \geq \Omega(n + \sqrt{n(\kappa - 1)}\ln\frac{1}{\epsilon}). \tag{B.2}$$

Some smaller low-accuracy terms are absorbed are ignored, as is done in [17]. For the case $n \geq \frac{1}{3}\kappa$, the proof of [17, Thm 2] tells us that, for any $k \geq 1$, there exist $L-$Lipschitz differentiable and $\mu-$strongly convex quadratic functions $f_1^k, f_2^k, ..., f_n^k$ and $F^k = \frac{1}{n}\sum_{i=1}^{n} f_i^k$, such that for any $x^0$, the $x^K$ produced after $K$ gradient evaluations, we have[15]

$$\mathbb{E}[F^K(x^K) - F^K(x^*)] \geq \frac{\mu}{4}\left(\frac{nR\mu}{L-\mu}\right)^2\left(\frac{\sqrt{1 + \frac{\kappa-1}{n}} - 1}{\sqrt{1 + \frac{\kappa-1}{n}} + 1}\right)^{\frac{2K}{n}},$$

where $R$ is a constant and $\kappa = \frac{L}{\mu}$.

Therefore, in order for $\epsilon \geq \mathbb{E}[F(x^k) - F(x^*)]$, we must have

$$\epsilon \geq \frac{\mu}{4}\left(\frac{nR\mu}{L-\mu}\right)^2\left(\frac{\sqrt{1 + \frac{\kappa-1}{n}} - 1}{\sqrt{1 + \frac{\kappa-1}{n}} + 1}\right)^{\frac{2K}{n}} = \frac{\mu}{4}\left(\frac{nR\mu}{L-\mu}\right)^2\left(1 - \frac{2}{1 + \sqrt{1 + \frac{\kappa-1}{n}}}\right)^{\frac{2k}{n}}.$$

Since $1 + \frac{1}{3}x \leq \sqrt{1 + x}$ when $0 \leq x \leq 3$, and $0 \leq \frac{\kappa-1}{n} \leq \frac{\kappa}{n} \leq 3$, we have

$$\epsilon \geq \frac{\mu}{4}\left(\frac{nR\mu}{L-\mu}\right)^2\left(1 - \frac{2}{2 + \frac{1}{3}\frac{\kappa-1}{n}}\right)^{\frac{2K}{n}},$$

or equivalently,

$$K \geq \frac{n}{2\ln(1+\frac{6n}{\kappa-1})}\ln\Big(\frac{\frac{\mu}{4}(\frac{nR}{\kappa-1})^2}{\epsilon}\Big).$$

As a result,

$$K \geq \frac{n}{2\ln(1+\frac{6n}{\kappa-1})}\ln\frac{1}{\epsilon} + \frac{n}{2\ln(1+\frac{6n}{\kappa-1})}\ln\Big(\frac{\mu}{4}(\frac{nR}{\kappa-1})^2\Big)$$

$$= \frac{n}{2\ln(1+\frac{6n}{\kappa-1})}\ln\frac{1}{\epsilon} + \frac{n}{2\ln(1+\frac{6n}{\kappa-1})}\ln(\frac{\mu R^2}{24}) + \frac{n}{\ln(1+\frac{6n}{\kappa-1})}\ln\frac{6n}{\kappa-1}.$$

Since $\frac{\ln\frac{6n}{\kappa-1}}{\ln(1+\frac{6n}{\kappa-1})} \geq \frac{\ln 2}{\ln 3}$ when $\frac{n}{\kappa-1} \geq \frac{n}{\kappa} \geq \frac{1}{3}$, for small $\epsilon$ we have

$$K \geq \frac{n}{2\ln(1+\frac{6n}{\kappa-1})}\ln\frac{1}{\epsilon} + \frac{n}{2\ln(1+\frac{6n}{\kappa-1})}\ln(\frac{\mu R^2}{24}) + \frac{\ln 2}{\ln 3}n$$

$$= \Omega\Big(\frac{n}{\ln(1+\frac{6n}{\kappa-1})}\ln\frac{1}{\epsilon}\Big) + \frac{\ln 2}{\ln 3}n \tag{B.3}$$

$$= \Omega\Big(\frac{n}{1+(\ln(n/\kappa))_+}\ln(1/\epsilon) + n\Big) \tag{B.4}$$

Now the expression in (B.4) is valid for $n \geq \frac{1}{3}\kappa$. When $n < \frac{1}{3}\kappa$, the lower bound in (B.4) is asymptotically equal to $\Omega(n\ln(1/\epsilon) + n)$, which is dominated by (B.2). Hence the lower bound in (B.4) is valid for all $\kappa, n$.

We may sum the lower bounds in (B.2) and (B.4) to obtain (B.1). This is because given an oblivious p-CLI algorithm, we may simply chose the adversarial example that has the corresponding greater lower bound. $\qquad\square$

## C   Lower Complexity Bound for SDCA

*Proof of Propsition 1.* Let $\phi_i(t) = \frac{1}{2}t^2$, $\lambda = \mu$, and $y_i$ be the $i$th column of $Y$, where $Y = c(n^2 I + J)$ and $J$ is the matrix with all elements being 1, and $c = (n^4 + 2n^2 + n)^{-1/2}(L-\mu)^{1/2}$. Then

$$f_i(x) = \frac{1}{2}(x^T y_i)^2 + \frac{1}{2}\mu\|x\|^2,$$

$$F(x) = \frac{1}{2n}\|Y^T x\|^2 + \frac{1}{2}\mu\|x\|^2,$$

$$D(\alpha) = \frac{1}{n\mu}\Big(\frac{1}{2n}\|Y\alpha\|^2 + \frac{1}{2}\mu\|\alpha\|^2\Big).$$

Since

$$\|y_i\|^2 = c^2\big((n^2+1)^2 + n - 1\big) = c^2(n^4 + 2n^2 + n) = L - \mu,$$

$f_i$ is $L$−smooth and $\mu$−strongly convex, and that $x^* = \mathbf{0}$.

We also have

$$\nabla D(\alpha) = \frac{1}{n\mu}\Big(\frac{1}{n}Y^2\alpha + \mu\alpha\Big) = \frac{1}{n\mu}\big((c^2 n^3 I + 2nc^2 J + c^2 J)\alpha + \mu\alpha\big),$$

So for every $k \geq 0$, minimizing with respect to $\alpha_{i_k}$ as in (2.5) yields the optimality condition:

$$0 = e_{i_k}^T \nabla D(\alpha^{k+1})$$

$$= \frac{1}{n\mu}\big(c^2 n^3 \alpha_{i_k}^{k+1} + 2c^2 n(\sum_{j\neq i_k}\alpha_j^k + \alpha_{i_k}^{k+1}) + c^2(\sum_{j\neq i_k}\alpha_j^k + \alpha_{i_k}^{k+1}) + \mu\alpha_{i_k}^{k+1}\big).$$

Therefore, rearranging yields:

$$\alpha_{i_k}^{k+1} = -\frac{(c^2 + 2c^2 n)}{c^2 n^3 + 2c^2 n + c^2 + \mu}\sum_{j\neq i_k}\alpha_j^k = -\frac{(c^2 + 2c^2 n)}{c^2 n^3 + 2c^2 n + c^2 + \mu}(e_{i_k}^T(J-I)\alpha^k).$$

As a result,

$$\alpha^{k+1} = (I - e_{i_k}e_{i_k}^T)\alpha^k - \frac{(c^2 + 2c^2 n)}{c^2 n^3 + 2c^2 n + c^2 + \mu}(e_{i_k}e_{i_k}^T(J - I)\alpha^k).$$

Taking full expectation on both sides gives

$$\mathbb{E}\alpha^{k+1} = \left((1 - \frac{1}{n})I - \frac{(c^2 + 2c^2 n)}{c^2 n^3 + 2c^2 n + c^2 + \mu}\frac{J - I}{n}\right)\mathbb{E}\alpha^k \triangleq T\mathbb{E}\alpha^k.$$

for linear operator $T$. Hence we have by Jensen's inequality:

$$\begin{aligned}
\mathbb{E}\big\|x^k\big\|^2 &= n^{-2}\mu^{-2}\mathbb{E}\big\|Y\alpha^k\big\|^2 \\
&\geq n^{-2}\mu^{-2}\big\|Y\mathbb{E}\alpha^k\big\|^2 \\
&= n^{-2}\mu^{-2}\big\|YT^k\alpha^0\big\|^2
\end{aligned}$$

We let $\alpha^0 = (1, \ldots, 1)$, which is an vector of $T$. Let us say the corresponding eigenvalue for $T$ is $\theta$:

$$\mathbb{E}\big\|x^k\big\|^2 \geq \theta^{2k}n^{-2}\mu^{-2}\big\|Y\alpha^0\big\|^2 \tag{C.1}$$
$$= \theta^{2k}\big\|x^0\big\|^2 \tag{C.2}$$

We now analyze the value of $\theta$:

$$\begin{aligned}
\theta &= (1 - \frac{1}{n}) - \frac{(c^2 + 2c^2 n)}{c^2 n^3 + 2c^2 n + c^2 + \mu}\frac{n-1}{n} \\
&= 1 - \frac{1}{n} - \frac{1 + 2n}{n^3 + 2n + 1 + \mu c^{-2}}\frac{n-1}{n} \\
&\geq 1 - \frac{1}{n} - \frac{1 + 2n}{n^3 + 2n + 1} \\
&\geq 1 - \frac{2}{n}
\end{aligned}$$

for $n > 2$. This in combination with (C.2) yields (2.7). □

## D  Nonconvex SVRG Analysis

*Proof of Theorem 3.* Without loss of generality, we can assume $x^* = \mathbf{0}$ and $F(x^*) = 0$.

According to lemma 3.3 and Lemma 5.1 of [20], for any $u \in \mathbb{R}^d$, and $\eta \leq \frac{1}{2}\min\left\{\frac{1}{L}, \frac{1}{\sqrt{mL}}\right\}$ we have

$$\mathbb{E}[F(x^{j+1}) - F(u))] \leq \mathbb{E}[-\frac{1}{4m\eta}\|x^{j+1} - x^j\|^2 + \frac{\langle x^j - x^{j+1}, x^j - u\rangle}{m\eta} - \frac{\mu}{4}\|x^{j+1} - u\|^2],$$

or equivalently,

$$\mathbb{E}[F(x^{j+1}) - F(u))] \leq \mathbb{E}[\frac{1}{4m\eta}\|x^{j+1} - x^j\|^2 + \frac{1}{2m\eta}\|x^j - u\|^2 - \frac{1}{2m\eta}\|x^{j+1} - u\|^2 - \frac{\mu}{4}\|x^{j+1} - u\|^2].$$

Setting $u = x^* = 0$ and $u = x^j$ yields the following two inequalities:

$$F(x^{j+1}) \leq \frac{1}{4m\eta}(\|x^{j+1} - x^j\|^2 + 2\|x^j\|^2 - 2(1 + \frac{1}{2}m\eta\mu)\|x^{j+1}\|^2), \tag{D.1}$$

$$F(x^{j+1}) - F(x^j) \leq -\frac{1}{4m\eta}(1 + m\eta\mu)\|x^{j+1} - x^j\|^2. \tag{D.2}$$

Define $\tau = \frac{1}{2}m\eta\mu$, multiply $(1 + 2\tau)$ to (D.1), then add it to (D.2) yields

$$2(1 + \tau)F(x^{j+1}) - F(x^j) \leq \frac{1}{2m\eta}(1 + 2\tau)\big(\|x^j\|^2 - (1 + \tau)\|x^{j+1}\|\big).$$

Multiplying both sides by $(1+\tau)^j$ gives

$$2(1+\tau)^{j+1}F(x^{j+1}) - (1+\tau)^j F(x^j) \le \frac{1}{2m\eta}(1+2\tau)\big((1+\tau)^j\|x^j\|^2 - (1+\tau)^{j+1}\|x^{j+1}\|\big).$$

Summing over $j = 0, 1, ..., k-1$, we have

$$(1+\tau)^k F(x^k) + \sum_{j=0}^{k-1}(1+\tau)^j F(x^j) - F(x^0) \le \frac{1}{2m\eta}(1+2\tau)(\|x^0\|^2 - (1+\tau)^k\|x^k\|^2).$$

Since $F(x^j) \ge 0$, we have

$$F(x^k)(1+\tau)^k \le F(x^0) + \frac{1}{2m\eta}(1+2\tau)\|x^0\|^2.$$

By the strong convex of $F$, we have $F(x^0) \ge \frac{\mu}{2}\|x^0\|^2$, therefore

$$F(x^k)(1+\tau)^k \le F(x^0)(2 + \frac{1}{2\tau}),$$

Finally, $\eta = \frac{1}{2}\min\{\frac{1}{L}, (\frac{1}{\overline{L}^2 m})^{\frac{1}{2}}\}$ gives

$$\frac{1}{\tau} = 4\max\{\frac{\kappa}{m}, (\frac{\overline{L}^2}{m\mu^2})^{\frac{1}{2}}\} \le 4(\frac{\kappa}{m} + (\frac{\overline{L}^2}{m\mu^2})^{-\frac{1}{2}}),$$

which yields

$$F(x^k) \le (1+\tau)^{-k}F(x^0)\big(2 + 2(\frac{\kappa}{m} + (\frac{\overline{L}^2}{m\mu^2})^{-\frac{1}{2}})\big).$$

To prove (4.2), we notice that

$$\tau = \frac{1}{4}\min\{\frac{m}{\kappa}, (\frac{m\mu^2}{\overline{L}^2})^{\frac{1}{2}}\},$$

so we have

$$\frac{1}{\ln(1+\tau)} \le \frac{1}{\ln(1+\frac{m}{4\kappa})} + \frac{1}{\ln\big(1 + (\frac{m\mu^2}{4\overline{L}})^{\frac{1}{2}}\big)}$$

Now for small $\epsilon$, the epoch complexity can be written as

$$K_0 = \lceil \frac{1}{\ln(1+\tau)}\ln\frac{F(x^0)(2+2(\frac{\kappa}{m} + (\frac{\overline{L}^2}{m\mu^2})^{-\frac{1}{2}}))}{\epsilon}\rceil$$

$$\le \mathcal{O}\Big((\frac{1}{\ln(1+\frac{m}{4\kappa})} + \frac{1}{\ln\big(1 + (\frac{m\mu^2}{4\overline{L}})^{\frac{1}{2}}\big)})\ln\frac{1}{\epsilon}\Big) + 1.$$

Since $m = \min\{2, n\}$, we have a gradient complexity of

$$K = (n+m)K_0 \le \mathcal{O}\Big((\frac{n}{\ln(1+\frac{n}{4\kappa})} + \frac{n}{\ln\big(1 + (\frac{n\mu^2}{4\overline{L}})^{\frac{1}{2}}\big)})\ln\frac{1}{\epsilon}\Big) + 2n.$$

And this is equivalent to the expression in (4.3).

$\square$

## Footnotes

[14]We absorb some smaller low-accuracy terms (high $\epsilon$) as is common practice. Exact lower bound expressions appear in the proof.

[15]note that for the SVRG in Algorithm 1 with $\psi = 0$, each update in line 7 is regarded as an iteration.