[Reviews · NeurIPS 2018]

Reviewer 1



Summary: The paper shows that SVRG and SARAH can be (modified to be) fundamentally faster than methods that satisfy the “span assumption”, which is followed by algorithms that minimize finite-sum smooth functions. Moreover, the author(s) also provide some lower bound results to show that the speedup is optimal when \kappa = O ( n ). Comments: The paper is well written and provides an interesting aspect for variance reduction methods. From my view, the results in this paper are non-trivial. I have the following comments: 1) When n >> \kappa, the author(s) show that SVRG and SARAH are faster than other variance reduction methods. According to the original SARAH paper, the author(s) discussed some advantages of SARAH over SVRG in the strongly convex case. Do you have any idea or intuition (in mind) about comparing these two methods together? 2) I saw that the paper is analyzing SVRG but also implies the same conclusion for SARAH. I somehow agree with the author(s) with this conclusion, but the analyses of SVRG and SARAH are totally different, especially SARAH has a biased estimator of gradient while SVRG has an unbiased one. Do you think whether it is trivial to apply the results of SVRG into SARAH? 3) I agree with the author(s) that the hybrid method(s) could be faster since the full gradient steps could ensure the iterations make some progress toward to the optimal solution. Section 3 is useful and I think it could help people have some intuitions in order to improve SVRG and SARAH in the future. 4) I understand that the paper focuses more on theoretical side. However, you could also consider to add some numerical experiments. It does not affect much, but it may make the paper become nicer. 5) A minor comment: The name of Section 4 is a little bit confused. I understand that you are considering the case where component functions f_i are not necessarily convex, but the objective function F is still strongly convex. The title may make the readers think that you also consider the nonconvex objective. In general, I think the paper is useful and could help people understand more about variance reduction methods.

Reviewer 2



Summary: The authors consider the finite-sum minimization problem. The work aims to investigate the “span condition” of modern variance reduced methods, and the role it plays in the complexity analysis of modern variance reduced methods. The span condition is satisfied for some method if the k-th iterate is formed as the starting iterate plus a linear combination of the stochastic gradients computed by the method. This condition is satisfied for many methods, including SAG, SDCA, SAGA, Finito and MISO. However, it is not satisfied for methods such as SVRG and SARAH. The authors ask the question whether the lack of the span condition in methods such as SVRG and SARAH might lead to better complexity estimates, both in terms of lower and upper bounds. They give positive answer to this question. In all results the authors assume all functions f_i to be smooth (with parameter L_i), and the average to be strongly convex (with parameter \mu). Specialized results are then proved in the case when the functions f_i are convex (CASE 1), and when they are not convex (CASE 2). Clarity: The paper is well written; the ideas are clearly introduced, motivated and explained. Quality/Originality/Significance: One of the key results of this paper (Theorem 2) is to show that in the regime when the condition number is not too large when compared to the number of functions (kappa = O(n)), the lower bound proved in [17] can be improved. This is done for CASE 1. The improvement is not very large, which is not surprising. In particular, in one expression, the term “n” is replaced by “n/(1+log(n/kappa))”. However, the relative improvement grows as n/kappa grows, and is clearly unbounded. The new lower bound is obtain by a modification/tightening of the bound obtained in [17]. Because of this, the tools used to obtain this result are not highly original. However, the result itself is quite interesting. The second key result (Theorem 1/Corollary 2) is a modified analysis of SVRG (used with a different stepsize) which matches the newly established lower bound (again, for CASE 1). The result itself can be derived from the original SVRG analysis by a different choice of the parameters of the method (stepsize, number of steps in the inner loop, number of outer loops). Hence, the result can be seen as a rather elementary observation. Still, it is interesting that this observation was not explicitly made before. Can you give examples of important problems or reasonable situations when one might expect n/kappa to be very large? This would strengthen the importance of the regime analyzed in this paper. The paper does not contain any experiments. I view this as a major issue and this omission weakens the paper considerably. I’d like to see the difference, through experiments, offered by the logarithmic speedup. It should be possible to test this and clearly illustrate the benefits as n/kappa grows, even if this is done on a carefully constructed synthetic problems. Various types of experiments can be done and should have been done. The authors provide a compelling explanation of why the “hybrid methods“ (i.e., methods such as SVRG that do not follow the span assumption) allow for this faster rate. This is done in Section 3. Additional contributions: a) Proposition 1: Provides a lower bound for SDCA (Omega(n log(1/eps))); which means that SDCA can not attain logarithmic speedup. Can SDCA be modified to allow for logarithmic speedup? b) Section 4: Provides logarithmic speedup for SVRG in CASE 2. However, I do not see a lower bound provided for this case. Can you establish a lower bound? Small issues: 1) Line 139: Missing right parenthesis 2) Line 139: “can can” -> “can” 3) Line 168: “X^*” -> “x^*” 4) Footnote 7: “sequence” -> “sequences” 5) Line 303: “shwartz” -> “Shwartz” Comment: My answer to Q4 should be ignored (it's a dummy response; one should really have a NA option) since there are no computational results to be reproduced. ----- POST AUTHOR FEEDBACK: I've read the rebuttal, and the other reviews. I am keeping the same score.

Reviewer 3



UPDATE after rebuttal: I still doubt the practical impact of the paper, but overall it is a nice contribution that closes a gap in theory. I still vote for acceptance. This paper considers methods for minimizing finite sum structured, strongly convex problems and shows that methods that compute a full gradient (i.e. do a full pass over the data from time to time, like in SVRG or SARAH) can be faster than methods without a full pass (for instance SAGA), given that the condition number of the problem is small (compared to the dimension n). The paper considers the SVRG variant and analysis from [21, Xiao Zhag] and makes the observation that one can obtain a slightly better estimate of the convergence rate when the condition number \kappa is small compared to the dimension n. This can result in a \log n speedup over previously known results in this parameter regime (i.e. the algorithm stays the same, just a more careful parameter setting leads to the faster rate). The main idea is as follows: stochastic algorithms that gradually improve the estimate of the solution (think for instance of coordinate descent) converge at a rate (1-1/n), i.e. by a constant factor each epoch. SVRG can converge by a factor (n/\kappa)^{1/2} instead, which is much better when n >> \kappa. The paper gives a lower bound for coordinate descent/SDCA, adapting the proof from [23, Arjevani] and a lower bound for SVRG (by slightly refining the proof from [17, Arjevani, Shamir]) that matches the upper bound. I think this paper is a nice theoretical contribution that settles some open questions on the complexity of SVRG. However, there is not clear practical impact. The improved rate can essentially only be obtained by knowing the condition number \kappa in advance. It is an open question whether there exists a parameter free algorithm that attains the same complexity. The paper distinguishes algorithms that satisfy the “span assumption” (like SGD) and algorithms that do not satisfy this assumption (like SVRG). However, to me the term “span assumption” sounds a bit misleading: the “trick” in all lower bound proofs based on this assumption is that the dimension n is much larger than the number of steps the algorithm performs. Clearly, algorithms that do a full pass over the data, like SVRG, do not fall into this class. But on the other hand, the iterates of SVRG also lie in the span of the first n gradients queried at x_0 and the subsequent randomly sampled gradients at iterates x_{n+i}. So, perhaps the result is more a consequence of the fact that the dimension is small (compared to the number of steps performed), and not on the fact that iterates are not contained in the span of some vectors? Maybe the authors could comment on that? There is typo on line 139.